# NON-METRIC MULTIDIMENSIONAL SCALING AT SCALE

## ABSTRACT

Multidimensional scaling (MDS) is a method to construct a low-dimensional embedding that approximates pairwise distances in a high-dimensional dataset. MDS exists in several flavors, with metric MDS approximating distances directly, while non-metric MDS additionally optimizing for an arbitrary monotonic transformation of the high-dimensional distances. Most existing MDS implementations have quadratic complexity and do not allow embedding large datasets; some fast stochastic MDS implementations have been recently developed, but only for metric MDS. Here we develop a fast MDS implementation, supporting both metric and non-metric MDS, using stochastic gradient descent in PyTorch. This allows us, for the first time, to construct non-metric MDS embeddings of datasets with sample sizes in tens of thousands. We conduct an empirical study of non-metric MDS using multiple simulated and real-world datasets, including a population genomic and a scRNA-seq dataset, and show that it can strongly outperform metric MDS in terms of global structure preservation.

## 1 INTRODUCTION

Multidimensional scaling (MDS) (Borg & Groenen, 2005) is one of the classic dimensionality reduction and embedding methods developed in the 1960s (Shepard, 1962a;b; Kruskal, 1964a;b). Its goal is to find low-dimensional embedding vectors $\mathbf{z}_i \in \mathbb{R}^p$ such that pairwise distances $\|\mathbf{z}_i - \mathbf{z}_j\|$ between them approximate some given pairwise distances $D_{ij}$. MDS is well-known in the field of dimensionality reduction (de Bodt et al., 2025) as the algorithm aiming to preserve global structure of the data, as opposed to various neighbor-embedding algorithms such as $t$-SNE (van der Maaten & Hinton, 2008) and UMAP (McInnes et al., 2018) that aim to preserve local structure.

Most existing implementations of MDS, such as the one in scikit-learn (Pedregosa et al., 2011) in Python or the `smacof` package in R (de Leeuw & Mair, 2009; Mair et al., 2022), have quadratic memory and runtime complexity, precluding its use for datasets with sample sizes $n \gg 10000$. Indeed, simply storing the $n \times n$ matrix of pairwise distances $\mathbf{D}$ in memory can already be prohibitive for large $n$.

Several stochastic versions of MDS have been developed in various contexts over the years (Agrafiotis, 2003; Zheng et al., 2018; Lambert et al., 2022) (Section 2). They have linear runtime complexity and allow to construct MDS embeddings of large datasets. All of them optimize so-called *metric MDS* where $\|\mathbf{z}_i - \mathbf{z}_j\|$ should directly approximate given $D_{ij}$. In a more flexible flavor of MDS, known as *non-metric MDS*, $\|\mathbf{z}_i - \mathbf{z}_j\|$ should approximate $D_{ij}$ only up to an arbitrary monotonic transformation, i.e. approximating only the order of distances and not the distances themselves. We are not aware of any fast implementations of non-metric MDS, or of any modern benchmark studies of dimensionality reduction methods that include non-metric MDS. Even though the method was originally developed in the 1960s, it seems to have largely faded out of the collective memory.

In this paper, we aim to fill this gap. Specifically,

- we fix multiple issues with the scikit-learn implementation of non-metric MDS;
- we develop a stochastic PyTorch implementation of both metric and non-metric MDS;
- we give examples of synthetic and real-world datasets where non-metric MDS strongly outperforms metric MDS in terms of common metrics of global embedding quality;
- we apply non-metric MDS to real-world transcriptomic and genomic datasets and argue that it can be useful as a visualization tool focusing on global structure preservation.

## 2   RELATED WORK

There have been multiple papers independently introducing and re-introducing stochastic metric MDS in various contexts. We are aware of three such works:

1. Lambert et al. (2022) developed SQuadMDS (stochastic quartet MDS) that optimizes metric MDS using mini-batches of size $b = 4$ (quartets). The authors developed an efficient C++ implementation and used it to obtain metric MDS embeddings of MNIST and the Tasic datasets, also analyzed here. Furthermore, the authors suggested a hybrid between SQuadMDS and $t$-SNE (van der Maaten & Hinton, 2008).

2. Zheng et al. (2018) developed a stochastic implementation of weighted MDS in the context of graph drawing. In that setting, the input matrix of distances is the matrix of shortest-path graph distances between pairs of graph nodes. For optimization, the authors sampled pairs of points, essentially using mini-batches of size $b = 2$.

3. Agrafiotis (2003) also developed a version of stochastic metric MDS with $b = 2$, calling it stochastic proximity embedding (SPE). This method was developed in the context of computational chemistry, to visualize large collections of chemical compounds (Agrafiotis et al., 2010).

At the same time, we are not aware of any existing scalable implementations of *non-metric* MDS. All examples in Borg & Groenen (2005) or Mair et al. (2022) have $n < 100$.

## 3   BACKGROUND

**Notation**   Let the high-dimensional dataset consist of $n$ vectors $\mathbf{x}_i \in \mathbb{R}^d$, assembled into a $n \times d$ data matrix $\mathbf{X}$. Let a $d \times d$ matrix $\mathbf{D}$ be a pairwise distance matrix with $D_{ij}$ being some distance (e.g. Euclidean, cosine, correlation, etc.) between $\mathbf{x}_i$ and $\mathbf{x}_j$. Alternatively, we may be given (e.g. experimentally obtain) some distance matrix $\mathbf{D}$ directly, without access to any $\mathbf{X}$, and without knowing the value of $d$.

The goal of any MDS algorithm is to construct low-dimensional embedding vectors $\mathbf{z}_i \in \mathbb{R}^p$ such that pairwise distances between them approximate pairwise distances $D_{ij}$. Typically $p = 2$ for visualization purposes. We refer the reader to Borg & Groenen (2005) for a detailed systematic treatment of MDS.

**Classical MDS**   In classical MDS, also known as Torgerson's scaling (Torgerson, 1952) and as principal coordinate analysis (PCoA), the assumption is that $\mathbf{D}$ is a matrix of Euclidean distances. If so, it can be converted into the Gram matrix of centered vectors $G_{ij} = (\mathbf{x}_i - \bar{\mathbf{x}})^\top (\mathbf{x}_j - \bar{\mathbf{x}})$ by double-centering the matrix of squared distances: $\mathbf{G} = -(\mathbf{I} - \mathbf{1}_n/n)(\mathbf{D} \odot \mathbf{D}/2)(\mathbf{I} - \mathbf{1}_n/n)$, where $\odot$ denotes elementwise multiplication, and $\mathbf{1}_n$ denotes the $n \times n$ matrix of all ones. If $\mathbf{G} = \mathbf{U}\mathbf{\Lambda}\mathbf{U}^\top$ is the eigendecomposition of $\mathbf{G}$, then $\mathbf{Z} = \mathbf{U}\mathbf{\Lambda}^{1/2}$ gives the principal components of $\mathbf{X}$ and the classical MDS of $\mathbf{D}$.

Note that if $\mathbf{D}$ is not a matrix of Euclidean distances, then classical MDS is not equivalent to PCA of $\mathbf{X}$. If $\mathbf{\Lambda}$ contains negative eigenvalues, then $\mathbf{D}$ cannot be obtained as Euclidean distances between vectors of any dimensionality.

**Metric MDS**   The loss function of metric MDS is called (raw) stress, and is simply the squared error between $D_{ij}$ and $\|\mathbf{z}_i - \mathbf{z}_j\|$:

$$\mathcal{L}_{\text{MMDS}} = \sum_{i<j} \left( \|\mathbf{z}_i - \mathbf{z}_j\| - D_{ij} \right)^2. \tag{1}$$

The common approach for minimizing this loss function relies on the iterative SMACOF algorithm (scaling by majorizing a complicated function) (de Leeuw, 1977). This algorithm is used in most common MDS implementations, e.g. in R (de Leeuw & Mair, 2009; Mair et al., 2022) and in the scikit-learn library (Pedregosa et al., 2011) in Python. SMACOF converges to a local minimum of $\mathcal{L}_{\text{MMDS}}$, but note that $\mathcal{L}_{\text{MMDS}}$ is not convex and has many local minima, so the final embedding depends on the initialization.

It is often more convenient to use some normalized version of stress, for example so-called stress-1 introduced by Kruskal (1964a;b):

$$\text{Stress-1} = \sqrt{\frac{\sum_{i<j} \left( \|\mathbf{z}_i - \mathbf{z}_j\| - D_{ij} \right)^2}{\sum_{i<j} \left( \|\mathbf{z}_i - \mathbf{z}_j\| \right)^2}}. \tag{2}$$

It can be shown that any minimum of $\mathcal{L}_{\text{MMDS}}$ corresponds to a minimum of stress-1, if $\mathbf{Z}$ is appropriately rescaled. The optimal scaling factor can be computed analytically, and the stress-1 of the rescaled embedding ends up being equal to $\sqrt{s^2/(1+s^2)}$ where $s$ is the stress-1 value of $\mathbf{Z}$ obtained by minimizing Eq. 1 (Borg & Groenen, 2005, Chapter 11.1). In the following, when talking about stress-1 of metric MDS, we will use this rescaled value.

**Non-metric MDS**    Non-metric MDS (Shepard, 1962a;b; Kruskal, 1964a;b) additionally allows an arbitrary monotonic transformation $f(\cdot)$ of the input distances:

$$\mathcal{L}_{\text{NMDS}} = \sqrt{\frac{\sum_{i<j} \left( \|\mathbf{z}_i - \mathbf{z}_j\| - f(D_{ij}) \right)^2}{\sum_{i<j} \left( \|\mathbf{z}_i - \mathbf{z}_j\| \right)^2}}. \tag{3}$$

Here, to avoid trivial solutions such as $f(\cdot) = 0$, the loss function needs to be normalized, which is why stress-1 is used as the loss. Fixing $f(\cdot)$ to identity recovers metric MDS (up to the scaling factor, see above).

The standard approach for minimizing this loss function is alternating minimization: after each step of the SMACOF algorithm, an optimal $f(\cdot)$ is fit by isotonic (monotonic) regression of current $\|\mathbf{z}_i - \mathbf{z}_j\|$ values on $D_{ij}$ values. For convenience, most implementations scale the resulting $f(D_{ij})$ values to have mean 1 (note that any such scaling does not affect the loss function, as $\mathbf{Z}$ can be scaled accordingly).

**Scikit-learn implementation and improvements for this paper**    The scikit-learn library (Pedregosa et al., 2011) in Python contained implementations of metric and non-metric MDS using the SMACOF algorithm. While working on this paper, we discovered a number of issues with that implementation and fixed them in a series of four PRs (PR numbers hidden to preserve anonymity), three of which have already been merged and the remaining one should be merged soon. Apart from fixing some bugs and implementation errors, we implemented the classical MDS algorithm and set it as default initialization for metric and non-metric MDS. All scikit-learn experiments reported below use our fork, which should be integrated into version 1.8. All experiments were run with default hyperparameters and the algorithm always reached the convergence criterion.

## 4   STOCHASTIC METRIC AND NON-METRIC MDS

**SGD implementation in PyTorch**    To implement a stochastic version of metric MDS, we split the dataset into $\lfloor n/b \rfloor$ mini-batches of size $b$ (randomly in each epoch). If the input data are given as $\mathbf{X}$ matrix, then for each mini-batch, we compute $b \times b$ values of $D_{ij}$ (in order to avoid storing the entire $n \times n$ matrix $\mathbf{D}$ in memory, as this can be prohibitive for large $n$). We normalize the $D_{ij}$ values to have unit mean in each mini-batch. Then we use Stress-1 (Eq. 2) as the loss, with both sums only running over the $b^2$ terms in the mini-batch.

The entire model is an instance of `torch.nn.Embedding`, initialized with PCA (in case $\mathbf{X}$ is used as input) or with classical MDS (in case $\mathbf{D}$ is used). We scale the initialization to have a pre-specified standard deviation $\sigma_{\text{init}}$ of the first component. For optimization we use Adam with learning rate $\lambda$ and cosine learning rate decay over $n_{\text{epochs}}$. Our choice to use cosine learning rate decay follows the common practice in the field, but also agrees with Agrafiotis (2003) and Zheng et al. (2018) who argued that starting with a large learning rate and decaying it down to zero helps optimizing stochastic metric MDS.

For the stochastic version of non-metric MDS, the only modification to the above is that $D_{ij}$ values are transformed with isotonic regression (we use `IsotonicRegression` from scikit-learn). As isotonic regression can be quite slow, we opted to fit it only on the first batch in each epoch; for subsequent batches in the same epoch, we use the fitted regression model to transform the $D_{ij}$ values. Then we use Stress-1 (Eq. 3) as the loss.

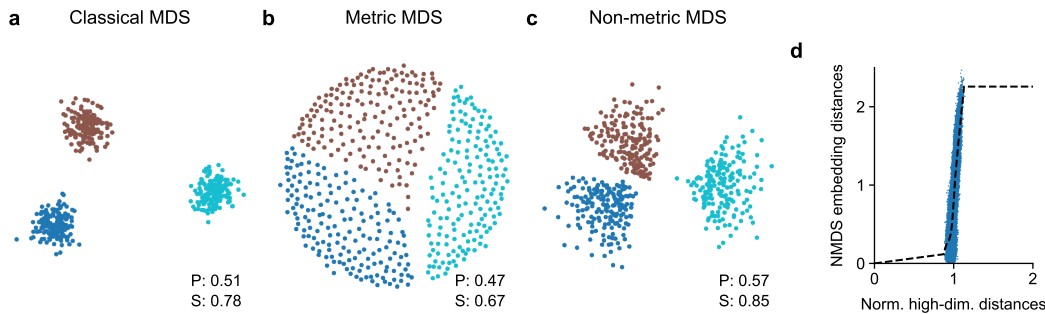

Figure 1: Classical, metric, and non-metric MDS embeddings of the noisy blobs dataset ($n = 500$ in 1002 dimensions). Values in the lower-right are Pearson and Spearman correlations between pairwise high-dimensional and low-dimensional distances. **(b–c)** Embeddings obtained using the SMACOF implementation in scikit-learn (with our improvements). **(d)** Shepard diagram for the non-metric MDS: embedding distances plotted against normalized high-dimensional distances. Dashed line shows monotonic regression fit.

**Default hyperparameters** As the default hyperparameters, we use $\lambda = 0.02$ and $\sigma_{\text{init}} = 0.5$ (which we found to work better than 1 on some of the datasets). For small datasets with $n < 5000$, we use batch size $b = 100$, and for larger datasets we use $b = 1000$. The default number of epochs is set to $n_{\text{epochs}} = 200$ if $n < 50000$ and to $n_{\text{epochs}} = 20$ otherwise. We found these parameters to work reasonably well across all considered datasets. We increased the number of epochs to $n_{\text{epochs}} = 500$ for non-metric MDS of the Tasic dataset, see below. Note that on some of our datasets a much smaller number of training epochs would suffice to reach a similar final loss.

**Runtime considerations** We observed that the optimal learning rate grew with the batch size (Figure S1) and that larger batch sizes were beneficial and allowed to achieve lower loss. On the other hand, the runtime of our implementation scales as $b^2 \cdot \lfloor n/b \rfloor \approx nb$, and in the non-metric case, the isotonic regression also becomes slower with larger $b$. We found $b = 1000$ to be a reasonable compromise.

All experiments reported in this paper were run on a consumer laptop with Intel Core i5 CPU at 1.6 GHz and 16 Gb RAM. The metric MDS could be accelerated by running our code on a GPU, but this would be more difficult for non-metric MDS, where isotonic regression can only run on the CPU and would need to be integrated into the data loader (or re-implemented for GPU). In this work, we did not aim to optimize for runtime.

## 5 EXPERIMENTAL RESULTS

### 5.1 NON-METRIC MDS SHOWS THE HIGHEST GLOBAL STRUCTURE PRESERVATION FOR VERY HIGH-DIMENSIONAL DATA

We started with analyzing a synthetic noisy blob dataset, with 500 points sampled from a mixture of three well-separated 2D Gaussians with additional 1000 non-discriminative Gaussian dimensions (Figure 1). PCA (or, equivalently, classical MDS) produced an embedding with well-separated classes (Figure 1a), whereas metric MDS could barely separate the classes (Figure 1b) and in fact typically failed to yield separated classes when using random initialization (by default we used classical MDS as initialization for metric MDS).

To evaluate the embeddings in a loss-independent way, we used Pearson and Spearman correlations between pairwise distances in the embedding space and in the original space, which is a commonly used metric to assess global embedding quality (Becht et al., 2019; Kobak & Berens, 2019; de Bodt et al., 2025). To compute the correlations for a given dataset, we randomly sampled 500 points out of $n$ and computed both correlation coefficients between $500^2$ pairwise distances $D_{ij}$ and embedding distances. We report both values, but prefer to use Spearman correlation as it is not affected by a monotonic transformation of $D_{ij}$ values.

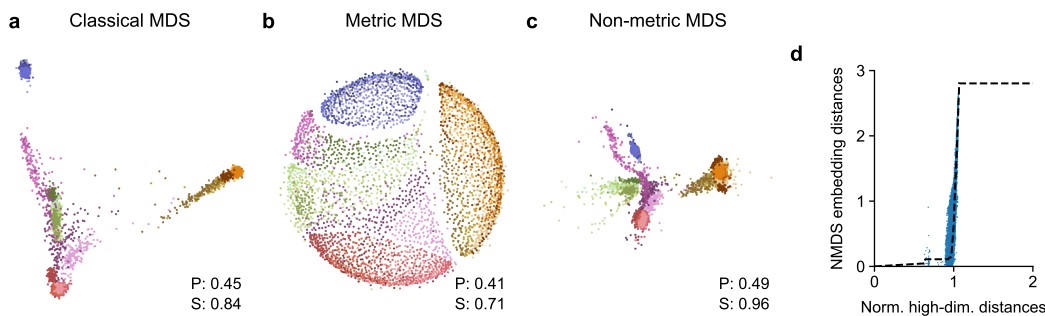

Figure 2: Classical, metric, and non-metric MDS embeddings of the 1000 Genomes dataset ($n = 3450$ in 53999 dimensions). See caption of Figure 1 for explanations. Colors correspond to the sampling population (26 global populations), see The 1000 Genomes Project Consortium (2015).

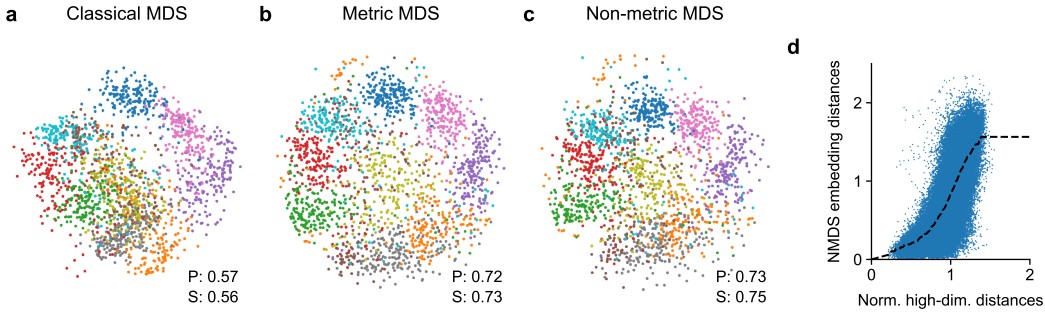

Figure 3: Classical, metric, and non-metric MDS embeddings of the Digits dataset ($n = 1797$ in 64 dimensions). See caption of Figure 1 for explanations.

On the noisy blob dataset, the Spearman correlation was higher for classical MDS (0.78) than for metric MDS (0.67). This is because the high-dimensional distances in this dataset are tightly concentrated (Figure 1d), making it impossible to reproduce them in two dimensions. Using the scikit-learn implementation of non-metric MDS with our improvements (see Section 3), we found that non-metric MDS resulted in a clearer class separation (Figure 1c) and the highest Spearman correlation (0.85). The isotonic regression transformed the highly-concentrated high-dimensional distances to a range between 0 and 2 (Figure 1d; in MDS literature this kind of plot is known as *Shepard diagram*).

We observed the same behavior in the 1000 Genomes dataset (The 1000 Genomes Project Consortium, 2015), analyzed in de Bodt et al. (2025) (we used the same preprocessing as there, resulting in 53999 feature dimensions) (Figure 2). Here input distances were also highly concentrated, metric MDS struggled to represent the data and produced a nearly-spherical embedding with Spearman correlation (0.71) lower than in PCA (0.84). In contrast, non-metric MDS produced a more informative embedding with better class separation and higher Spearman correlation (0.96).

On the other hand, on the Digits dataset (available in scikit-learn) there was almost no difference between the performance of metric and non-metric MDS (Figure 3).

## 5.2 STOCHASTIC MDS IMPLEMENTATION IS AS GOOD AS SMACOF BUT MUCH FASTER

Using all three datasets presented above (noisy blobs, 1000 Genomes, and Digits), we compared scikit-learn implementation (with our modifications) with our stochastic implementation. We found that both implementations achieved similar stress-1 values in most cases (Table 1). In 5 out of 6 cases there was no difference, and in one case (non-metric MDS of 1000 Genomes dataset) scikit-learn achieved 6% lower loss, but visually an almost identical embedding.

At the same time, the stochastic implementation was much faster in all cases with $n > 1000$ (Table 2). The speed-up was particularly noticeable for non-metric MDS.

Table 1: Stress-1 values obtained using the SMACOF and the SGD implementations. On all datasets, our stochastic implementation achieves the same embedding quality as the reference non-stochastic implementation. Tasic and MNIST datasets are too large for the non-stochastic implementation.

|  | Blobs | Genomes | Digits | Tasic | MNIST |
|---|---|---|---|---|---|
| Metric MDS, sklearn | 0.401 | 0.406 | 0.328 | – | – |
| Metric MDS, stochastic | 0.402 | 0.408 | 0.329 | 0.247 | 0.359 |
| Non-metric MDS, sklearn | 0.254 | 0.129 | 0.292 | – | – |
| Non-metric MDS, stochastic | 0.247 | 0.137 | 0.283 | 0.095 | 0.293 |

Table 2: Runtimes using the SMACOF and the SGD implementations. The stochastic implementation is faster for all datasets with $n > 1000$. Tasic and MNIST datasets are too large for the non-stochastic implementation. The runtimes correspond to the hardware and hyperparameters described in the text.

|  | Blobs | Genomes | Digits | Tasic | MNIST |
|---|---|---|---|---|---|
| Sample size ($n$) | 500 | 3450 | 1797 | 23822 | 70000 |
| Metric MDS, sklearn | 1 s | 72 s | 9 s | – | – |
| Metric MDS, stochastic | 5 s | 15 s | 4 s | 4 m | 14 m |
| Non-metric MDS, sklearn | 3 s | 204 s | 131 s | – | – |
| Non-metric MDS, stochastic | 6 s | 20 s | 6 s | 37 m | 17 m |

The scikit-learn implementation constructs the full $\mathbf{D}$ matrix in memory, which becomes prohibitive for $n \gg 10000$. Due to memory overflow, we were unable to run it on our two further datasets: the Tasic dataset of single-cell transcriptomic data from mouse cortex (Tasic et al., 2018), analyzed previously in Kobak & Berens (2019) and de Bodt et al. (2025) (we followed the preprocessing from these two papers), and the MNIST dataset. So for these two datasets, all following results were obtained using our stochastic implementation.

On the Tasic dataset, non-metric MDS produced an embedding with the highest Spearman correlation (0.96, vs. 0.91–0.92 for PCA and metric MDS) (Figure 4). One salient feature of this dataset is that it contains a small amount of various non-neural cells that are (a) far away from all neural cells; and (b) far away from each other. Non-metric MDS emphasized this structure by locating non-neural cells along an arc going around the cluster of neural cells. This happened at a price of squeezing all neural cells together, compared to metric MDS embedding. Note that we increased the $n_{\text{epochs}}$ value for non-metric MDS here to 500; our default value of 200 produced a similar-looking embedding but with less pronounced 'arc' structure.

On the MNIST dataset, the difference between metric and non-metric MDS was visually not very apparent (Figure 5), but non-metric MDS did show higher global structure preservation as measured by the Spearman correlation (0.80 vs. 0.65).

Across all five datasets analyzed in this paper, non-metric MDS achieved the highest Pearson and Spearman correlations in 7 out of 10 cases (five datasets and two evaluation metrics), and a shared highest in the remaining 3 cases (Tables 3 and S1).

## 6 DISCUSSION

Non-metric MDS was introduced over 60 years ago (Shepard, 1962a;b; Kruskal, 1964a;b) but has received almost no attention in modern manifold-learning and dimensionality-reduction research literature, compared to metric MDS (de Bodt et al., 2025). This is likely at least partially due to the lack of scalable implementations. Here we introduced a simple stochastic implementation of both metric and non-metric MDS. Furthermore, we showed that on some datasets, in particular very high-dimensional datasets with high levels of high-dimensional noise, non-metric MDS can produce

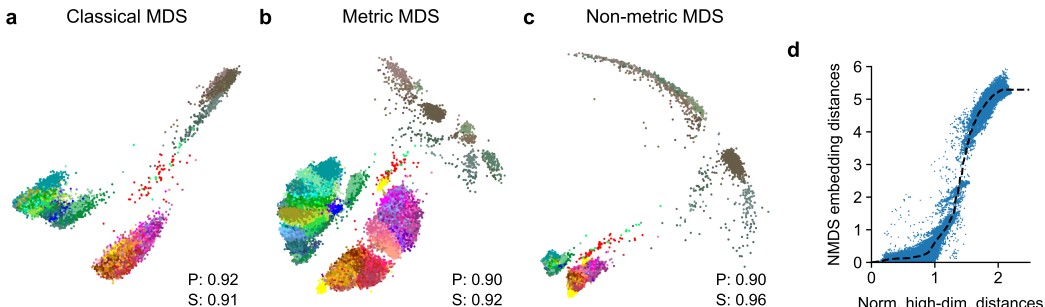

Figure 4: Classical, metric, and non-metric MDS embeddings of the Tasic dataset ($n = 23822$ in 50 dimensions). See caption of Figure 1 for explanations. Colors correspond to cell types (warm colors: inhibitory neurons; cold colors: excitatory neurons; brown/grey colors: non-neural cells). See Figure S2 for the same Shepard diagram as in (d), but colored by cell identity of cell pairs.

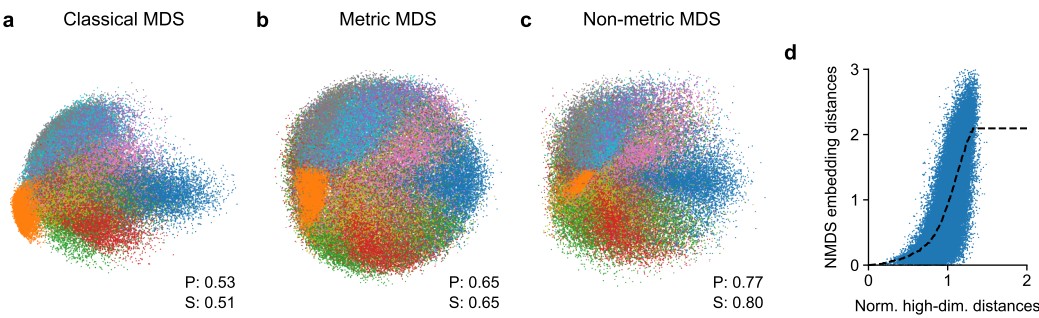

Figure 5: Classical, metric, and non-metric MDS embeddings of the MNIST dataset ($n = 70000$ in 784 dimensions). See caption of Figure 1 for explanations.

a very different kind of embedding compared to classical MDS and metric MDS, and can strongly outperform them in terms of global structure preservation.

Several recent studies have developed hybrid methods (Lambert et al., 2022; Kury et al., 2025) combining neighbor-embedding methods such as $t$-SNE (van der Maaten & Hinton, 2008) or UMAP (McInnes et al., 2018) that exhibit high local structure preservation with global methods such as PCA or MDS. Our results suggest that non-metric MDS can be an interesting choice for a global counterpart in such hybrid methods.

Overall, we hope that our paper will re-introduce non-metric MDS into the realm of modern dimensionality reduction methods.

Table 3: Spearman correlation coefficients between pairwise distances in the high-dimensional space and in the embedding. The highest value in each column is highlighted in bold, together with all values within 0.02. Non-metric MDS always achieves the best (or one of the best) result(s). See Table S1 for Pearson correlations.

|  | Blobs | Genomes | Digits | Tasic | MNIST |
|---|---|---|---|---|---|
| Classical MDS | 0.78 | 0.84 | 0.56 | 0.91 | 0.51 |
| Metric MDS | 0.67 | 0.71 | **0.73** | 0.92 | 0.65 |
| Non-metric MDS | **0.85** | **0.96** | **0.75** | **0.96** | **0.80** |

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

# A  SUPPLEMENTARY FIGURES AND TABLES

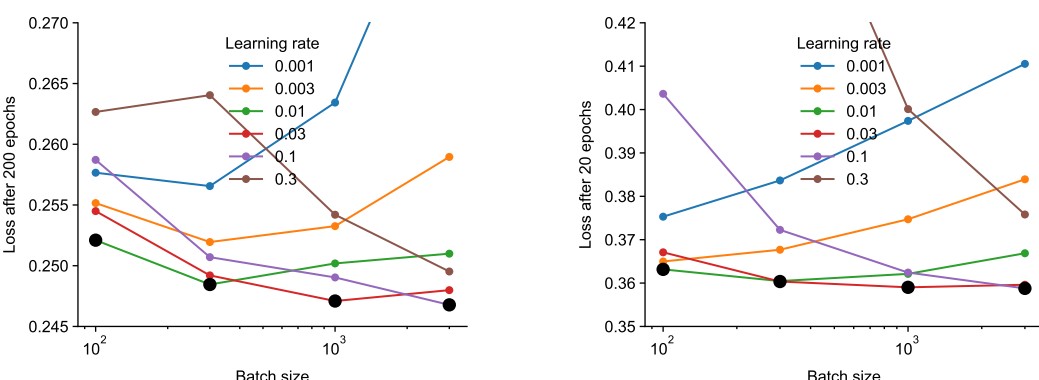

Figure S1: Stress-1 obtained using the SGD implementation of metric MDS using different batch sizes and learning rates for the Tasic (left; 200 epochs) and MNIST (right; 20 epochs) datasets. It is beneficial to use larger batch sizes together with higher learning rates. The runtime scales linearly with the batch size, see text.

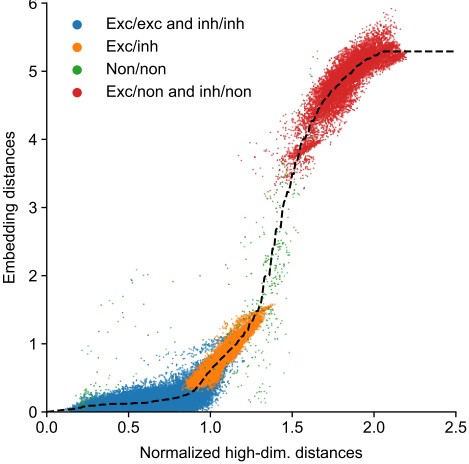

Figure S2: Shepard diagram for the non-metric MDS embedding of the Tasic dataset. Reproduced from Figure 4d but here each pair of cells is colored based on the cells' identities: excitatory/excitatory and inhibitory/inhibitory neuron pairs are shown in blue; excitatory/inhibitory neuron pairs — in orange; non-neural/non-neural pairs — in green; neural/non-neural pairs — in red.

Table S1: Pearson correlation coefficients between pairwise distances in the high-dimensional space and in the embedding. The highest value in each column is highlighted in bold, together with all values within 0.02. Non-metric MDS always achieves the best (or one of the best) result(s).

|                | Blobs | Genomes | Digits | Tasic | MNIST |
|----------------|-------|---------|--------|-------|-------|
| Classical MDS  | 0.51  | 0.45    | 0.57   | **0.92** | 0.53  |
| Metric MDS     | 0.47  | 0.41    | **0.72** | **0.90** | 0.65  |
| Non-metric MDS | **0.57** | **0.49** | **0.73** | **0.90** | **0.77** |