# OpenReview forum: "Non-metric multidimensional scaling at scale"
_ICLR.cc/2026/Conference — ICLR 2026 Conference Withdrawn Submission_

### Official Review · Reviewer_Gvcu · 2025-10-27

**Soundness:** 2
**Presentation:** 2
**Contribution:** 2
**Rating:** 2
**Confidence:** 4

**Summary:**

The paper presents a scalable implementation of both metric and non-metric multidimensional scaling (MDS) using SGD. It improves upon existing SMACOF-based methods by enabling non-metric MDS for large dataset. The authors show that non-metric MDS often preserves global structure better than metric MDS or PCA, especially for high-dimensional noisy data.

**Strengths:**

- The motivation is sound and I believe a scalable implementation benefits the community a lot in practice.

- The empirical improvements are good, for stress minimization.

**Weaknesses:**

I think it is really nice to have this implementation, but I regret to say the technical depth does not reach a certain bar. The main contribution, summarized in Section 4, is basically about implementation details on parameters and functions. I think this is at a course project level.

I would like to point out related works on non-Euclidean MDS. [1] still considers metric space, but [2] follows up on [1] and considers non-metric setting. The algorithms there, are combinatorial and do not use SGD etc. These methods, or at least [2], should be compared with your proposed implementation.

[1] How can classical multidimensional scaling go wrong?
[2] Neuc-MDS: Non-Euclidean Multidimensional Scaling Through Bilinear Forms

[2] is optimal to minimize a lower bound of Stress (not stress exactly). And I think it is important to question, if we can have optimal linear algorithms, why do we have to use SGD? Is the runtime improvement significant enough? Is the outcome always stable?

**Questions:**

See above

---

> ### Author Response · Authors · 2025-11-17
>
> Thank you for your review. As none of the four reviewers liked the paper, we are going to withdraw it and resubmit elsewhere. Still, here are some brief replies.
>
> We agree that "the technical depth" of our implementation is not very high. Still, nobody has ever seen a non-metric MDS embedding of any datasets commonly used in recent benchmarks of dimensionality reduction methods. We are literally the first to study this, and we give examples of both simulated and real-life datasets where non-metric MDS strongly outperforms MDS in terms of established global quality metrics! We are somewhat surprised that none of the reviewers found this aspect interesting.
>
> Thank you for bringing up these two references. We will include them into revision. Note that both these papers [1] and [2] study _classical_ MDS and specifically focus on properties of embedding into high-dimensional embedding spaces. In contrast, we study _non-metric_ MDS and specifically focus on two-dimensional embeddings. The paper [2] does develop "non-Euclidean" MDS, but it is still a version of classical MDS (obtained via the eigendecomposition of the double-centered distance matrix), and it is unrelated to our work on _non-metric_ MDS. These papers are not aimed at 2D visualization and do not contain a single 2D embedding. Anyway, we are of course happy to cite them in our Background section on classical MDS.

---

### Official Review · Reviewer_Sang · 2025-10-28

**Soundness:** 1
**Presentation:** 2
**Contribution:** 1
**Rating:** 0
**Confidence:** 5

**Summary:**

The paper proposes to compute metric and non-metric multidimensional scaling through stochastic gradient descent on its original loss function. They compare their approach to an sklearn implementation of MDS regarding a single metric.

**Strengths:**

- The paper lacks any originality, quality and significance (see Weaknesses). I am sorry for this harsh statement, but selling the application of SGD to an existing loss of an existing (old) method as a contribution is one of the more audacious things I have experienced as a reviewer.

**Weaknesses:**

-	The paper presents a solution to “scale” MDS that most people use anyway without making a paper out of it: stochastic gradient descent optimizing the vanilla MDS objective. I find it frankly audacious to submit this typical undergrad ML course question to a top conference. This approach **lacks novelty and relevance**.
-	The paper makes several strong claims regarding the relevance and scope of MDS in modern research. MDS has **not** “faded out of collective memory” as the authors suggest. In fact, it is regularly compared to in state of the art methodology (which the authors largely ignore), where **it performs much worse on relevant metrics as it fails to capture the usually intrinsically low-dimensional structure of the data**.
-	In more details regarding evaluation, this paper **lacks comparisons to any state of the art low-dimensional embedding method** including tSNE, UMAP, or the more recent LargeVis [1] and NCVis [2] or DensMAP[3]. Furthermore, it **lacks the usual benchmark metrics for evaluating low-dimensional embeddings**, such as local reconstruction quality (e.g. based on k-nearest neighbor), reconstruction of local scales (e.g. based on comparing local distances between each other), and ability to perform classification of relevant labels (e.g. kNN classifier on cell types in single-cell data). See e.g. [4] for such metrics.
-	The related work does not discuss any relevant recent work in the field of estimating low-dimensional embeddings beyond their particular approach. Apart from the methods mentioned above, there have been interesting recent developments to learn embeddings in a low-dimensional hyperbolic space [5], through angle preservation [6], or reconstructing local distances at multiple scales [7] each having their own advantages depending on considered data.

[1] Tang, J et al. *Visualizing Large-scale and High-dimensional Data.* WWW 2016.

[2] Artemenkov, A, Panov, M. *Ncvis: Noise contrastive approach for scalable visual-
Ization.* WWW 2020.

[3] Narayan, A, et al. *Assessing single-cell transcriptomic variability through density-preserving data visualization.* Nature Biotechnology, 39(6):765–774, 2021.

[4] Kobak, D, Berens, P. *The art of using t-sne for single-cell transcriptomics.* Nature Communications, 10(1):1–14, 2019.

[5] Keller-Ressel, M, Nargang, S, *Hydra: a method for strain-minimizing hyperbolic embedding of network- and distance-based data.* Journal of Complex Networks, 8(1):cnaa002, 02 2020.

[6] Fischer, J, Ma, R, *Sailing in high-dimensional spaces: Low-dimensional embeddings through angle preservation*, arXiv:2406.09876, 2024.

[7] Kury, N et al., *DREAMS: Preserving both Local and Global Structure in Dimensionality Reduction*, arXiv:2508.13747, 2025.

**Questions:**

-	What is the **novelty** in this paper?
-	How does the approach compare to **state-of-the-art** approaches for estimating low-dimensional embeddings on **established benchmark metrics**?
-	How does this work **fit it to the contemporary literature** of this topic?

---

> ### Author Response · Authors · 2025-11-17
>
> Thank you for your review. As none of the four reviewers liked the paper, we are going to withdraw it and resubmit elsewhere. Still, here are some brief replies.
>
> Your stated confidence is 5, and at the same time you seem to have missed that we are studying **non-metric** MDS! For example, you say that MDS is regularly used for comparisons in recent benchmarks. But this is only true for metric MDS! Can you point us to a single example of a modern dimensionality reduction paper that uses _non-metric_ MDS for comparisons? We are not aware of any such examples.
>
> Contrary to what you say, MDS does not "perform much worse" than t-SNE/UMAP/etc in all respects. Of course t-SNE/UMAP are much better in terms of local quality. But many modern papers, including for example [4] and [7] that you cited, use the same global metric (rank correlation between pairwise distances) that we use! Our metric **is** an "established benchmark metric"! And all these papers find that metric MDS often performs the best in terms of this metric. And we show that _non-metric_ MDS can on some datasets perform much better still! Isn't it at least a little interesting? "The paper lacks any originality, quality and significance" -- come on! For example [7] uses MDS in a hybrid method (DREAMS) interpolating between t-SNE and MDS. Using our non-metric MDS in their setup would lead to higher DREAMS score! We will make it clearer in the revision.
>
> We did not include comparisons to t-SNE/UMAP because we thought the outcome of such comparisons is obvious. Of course t-SNE/UMAP will strongly outperform MDS in terms of local metrics like kNN recall and kNN accuracy. At the same time, t-SNE/UMAP will **strongly lose** to MDS in terms of global metrics like rank correlation of pairwise distances (our Table 3). Maybe indeed it is a good idea to add t-SNE/UMAP to Table 3, but of course they will strongly lose! We can also add some discussion about t-SNE/UMAP preserving local structure much better.

---

> > ### Comment · Reviewer_Sang · 2025-11-22
> > **Answer to rebuttal**
> >
> > - **non-metric MDS.** Indeed you work on non-metric MDS, but that has nothing to do with my original point. You were claiming that MDS “faded out of collective memory” and I was countering that.
> >
> > - **MDS does not "perform much worse" than t-SNE/UMAP/etc**. Show it. You just claim something without providing scientific evidence.
> >
> > - **Our metric is an "established benchmark metric".** Yes, but it is only **one** of them, where people usually compare across several to show the different trade-offs a method brings.
> >
> > - **Using our non-metric MDS in their setup would lead to higher DREAMS score.** Again, this is a claim without scientific evidence. You hypothesize it improves DREAMS, but you can not simply say it does improve.
> >
> > - **We did not include comparisons to t-SNE/UMAP because we thought the outcome of such comparisons is obvious.** Well, that is exactly my point. They are performing better. You seem to **miss a central point of the last decade of research in the field**: The reason why methods like tSNE, UMAP and others were invented and the interest has shifted to such approaches is that **complex, high-dimensional data usually lies on an intrinsic, low-dimensional manifold**. In other words, long-range distances are almost meaningless to understand the structure of the data.
> >
> > I will stay with my score.

---

> > > ### Author Response · Authors · 2025-11-25
> > >
> > > OK thanks. I am going to take your valuable opinion into account when revising the paper for another venue. As I am retracting it now, the paper will get de-anonymized, so you will see that I am a co-author on two of the papers you listed as the ones I am ignoring :-) Anyway, I will try to convey the aims of this paper better when submitting elsewhere, and include some comparisons to t-SNE/UMAP.

---

### Official Review · Reviewer_xtTt · 2025-11-01

**Soundness:** 2
**Presentation:** 1
**Contribution:** 1
**Rating:** 2
**Confidence:** 5

**Summary:**

Multi-dimensional Scaling (MDS) is a classical dimensionality reduction algorithm that approximates pairwise distances. The authors proposed a faster implementation for both metric and non-metric MDS with PyTorch, and performed empirical study of the performance of these implementations on five datasets.

**Strengths:**

The authors re-discovered the non-metric MDS, from the 1960s, and wrote a modern implementation for the algorithm.

**Weaknesses:**

- **Code and related assets are not provided.** Given that two of the major contributions claimed by authors (Sec 1, L048-053) are implementations of algorithms under the MDS family, it's **very, very hard** to justify the decision. It's hard for me to understand the actual contribution of this paper. The author could perform proper anonymization and share the result. In the end, I tried to understand the implementation from the description in section 4, and I hardly believe that the effort here matches the effort typically required at a top-tier machine learning conference.

- **Insufficient comparison.** The author claimed the significance of the work to be the re-implementation of the non-metric MDS, yet there's limited comparison against other more recent dimensionality reduction algorithms. The only comparisons listed are against the classical and metric MDS, and no recent DR algorithms are compared against. Qualitative results shown in the paper are also bad: taking the well-known MNIST as an example, the visualization proposed in figure 5 is very problematic, as it mixes all the categories together. To some point, it's even worse than the bare-metal MDS shown in the figure. I believe modern algorithms, such as t-SNE and UMAP, can behave much, much better than the proposed algorithm in terms of structure preservation.

- **Limited contribution**. The author only aims to improve the implementation of the MDS. In the end, on line 196, the author refused to implement a dataloader for the algorithm and choose not to optimize for the runtime, without much explanation. The limited concern on run time and effort ultimately limits the contribution of the proposed implementation. The algorithm is way too slow -- modern algorithms can easily accomplish the MNIST algorithm within 1-2 minutes. I failed to see how the proposed algorithm could help advance the dimensionality reduction field in general or contribute to further downstream scientific contributions.

- Minor
  - L78: shouldn't the dimension of the matrix be n x n?

**Questions:**

See weaknesses for a plethora of questions.
- The authors claimed that the classical, metric and non-metric MDS algorithms are all variants of the MDS algorithm. Yet, as shown in figure 1, 2 & 4, their outputs on the simple dataset are very, very different, to the point that one could believe the algorithm to be completely different. This is not common in dimensionality reduction at all -- the difference between the parametric and non-parametric UMAP is much smaller. I'd like to know 1) why the behavior is so different and 2) if the difference is inherent to the algorithms, whether we should categorize them as the same set of algorithms.

**Details Of Ethics Concerns:**

- Potential double-blind violation on line 136. Revealing certain implementation is going to be merged in PRs could reveal the authors' identity.

---

> ### Author Response · Authors · 2025-11-17
>
> Thank you for your review. As none of the four reviewers liked the paper, we are going to withdraw it and resubmit elsewhere. Still, here are some brief replies.
>
> We did not include comparisons to t-SNE/UMAP because we thought the outcome of such comparisons is obvious. Of course t-SNE/UMAP will strongly outperform MDS in terms of local metrics like kNN recall and kNN accuracy. At the same time, t-SNE/UMAP will strongly lose to MDS in terms of global metrics like rank correlation of pairwise distances (our Table 3). Maybe indeed it is a good idea to add t-SNE/UMAP to Table 3, but of course they will strongly lose! We can also add some discussion about t-SNE/UMAP preserving local structure much better.
>
> You are right that the non-metric MDS embedding of MNIST is disappointing in comparison to t-SNE/UMAP, and runs slower t-SNE/UMAP. However, this is also a result of our paper! _Nobody has ever seen_ non-metric MDS embedding of MNIST, isn't it interesting to look at it? For the Tasic and the Genomes datasets, non-metric MDS does produce interesting embeddings, complementing t-SNE/UMAP.
>
> Regarding the naming: you are right that classical, metric, and non-metric MDS on some of our datasets look very differently! We did not come up with the names, they exist since the 1960s. But one of our main contributions is exactly this: to point out situations (both simulated and real-life datasets), where non-metric MDS is very different from metric MDS, and obtains better results. We are somewhat surprised that none of the reviewers found this aspect interesting.
>
> PS. Regarding ethics concern: most ICLR submissions are available on arXiv, so a simple Google search would reaveal authors' identities. This is not considered a double-blind violation. We did mention our PRs to scikit-learn, but to find them, one would need to search for them on purpose. We think this is not a double-blind violation either.

---

> > ### Comment · Reviewer_xtTt · 2025-11-24
> >
> > Unfortunately I don't think the rebuttal convinced me at all in asserting the value of this paper, or even the authors' willingness to improve it. If the proposed method is "disappointing in comparison to t-SNE/UMAP" when it comes to embedding generation, its value would also be disappointing. The authors failed to show how the proposed method is better than the existing method on Tasic / Genome dataset, and did not even try to accomplish such a comparison. Running t-SNE/UMAP on either one of them takes less than a few minutes, yet such comparison does not appear in this paper. Again, as all reviewers have pointed out, the amount of effort involved in this paper fell far below the bar of ICLR, or any top-tier machine learning conference. I will keep my score.

---

> > > ### Author Response · Authors · 2025-11-25
> > >
> > > OK thanks. I will take your criticism into account when revising the paper for another venue.

---

### Official Review · Reviewer_EtK3 · 2025-11-01

**Soundness:** 2
**Presentation:** 2
**Contribution:** 2
**Rating:** 2
**Confidence:** 4

**Summary:**

### Summary
This paper proposes a scalable PyTorch-based implementation of both metric and non-metric multidimensional scaling (MDS) using stochastic mini-batch optimization.

* Metric MDS: Minimizes the Stress-1 objective on mini-batches and directly optimizes the embedding $Z$ using Adam (not via a neural network). Initialization is performed with PCA, and each batch’s distance matrix $D_{ij}$ is normalized to have a mean of 1.
* Non-metric MDS: Builds upon the metric MDS algorithm by additionally applying isotonic regression. At the start of each epoch, an isotonic regression learns a monotonic transformation $f(\cdot)$; subsequent batches then optimize against $f(D_{ij})$ using the same procedure as in the metric case.
* Implementation: Implemented in PyTorch, storing $Z$ with `nn.Embedding` and avoiding full distance-matrix storage.
* Evaluation: Experiments on synthetic datasets, Digits, 1000 Genomes, Tasic, and MNIST demonstrate that non-metric MDS better preserves global (rank-based) structure. The paper also reports improvements to scikit-learn’s MDS implementation.

### Contributions

The paper’s main contribution lies in its engineering and scalability demonstration. The integration of Stress-1 minimization, mini-batch optimization, Adam, and isotonic regression enables practical large-scale non-metric MDS, which has clear practical value.
However, the theoretical contribution appears limited, and the motivation for several design choices (e.g., normalization, optimization setup) as well as ablation studies or formal analysis is not sufficiently elaborated. Overall, the work would require stronger theoretical justification and analysis to meet ICLR’s standards for originality and rigor.

**Strengths:**

* Demonstrates scalable non-metric MDS with clear practical value.
* Confirms the advantage of non-metric MDS in preserving global rank structures across multiple datasets.
* Includes community contribution through improvements to scikit-learn.

**Weaknesses:**

1. Limited theoretical contribution
   The approach primarily integrates existing components (Stress-1, Adam, isotonic regression) rather than introducing new theoretical insights or optimization principles.

2. Unclear distinction from existing stochastic MDS methods
   The differences from prior stochastic MDS variants—such as batch normalization to mean 1, the use of Stress-1, or Adam optimization—are not clearly justified either theoretically or empirically. The paper would benefit from clearer motivation and ablation analyses.

3. Inaccurate and incomplete complexity discussion

   * The paper describes most existing methods as having quadratic complexity, which appears to be inaccurate. The SMACOF algorithm involves pseudo-inverse computations that are $O(n^3)$ in the number of data points $n$, which dominate runtime.
   * The stochastic version of MDS is described as “linear,” but it is not clear with respect to which variable. Typically, one iteration costs $O(bd)$–$O(bd +\text{distance computations})$, where $b$ is the number of batch pairs and $d$ is the embedding dimension; over $T$ iterations, this scales as $O(Tbd)$. The authors should clarify the reference variable for “linear” and present a precise Big-$O$ analysis.
   * The non-metric optimization method combines the metric solver with isotonic regression at each epoch; therefore, its asymptotic computational cost should be at least as high as that of the metric MDS.
   * PyTorch defaults to fp32, whereas CPU-based solvers often use fp64, which can affect convergence stability and speed. It would be helpful if the paper explicitly reported precision to ensure fair comparison.
   * Experimental results should also report the mean and standard deviation across multiple runs to provide a fair and statistically reliable evaluation.

4. Lack of comparison with recent related work
   Recent studies have explored stabilized and accelerated MDS solvers. For instance,

   Fang, Z., Su, X., Tabuchi, T., Huang, J., and Kasai, H. (2025). *StableMDS: A Novel Gradient Descent-Based Method for Stabilizing and Accelerating Weighted Multidimensional Scaling.* AISTATS 2025

   proposes a gradient-descent-based MDS method with improved stability and convergence speed.
   Although the authors’ paper focuses on non-metric MDS, its optimization still depends on a metric solver. Therefore, a comparison with the latest research and other recent accelerated MDS methods (in terms of computational complexity, stability, and scaling behavior) would help better position this work within the context of recent MDS research, especially since the paper already compares against SMACOF.

5. Missing algorithmic pseudocode
   Including pseudocode illustrating the training flow (e.g., isotonic regression timing, batch updates, and distance computation strategy) would significantly improve clarity and reproducibility.

**Questions:**

1. Can you provide a formal Big-$O$ complexity analysis, explicitly accounting for the $O(n^3)$ term in SMACOF and describing how your method scales with $b$, $d$, and $T$?
2. What numerical precision (fp32 or fp64) was used, and how do these choices affect runtime and convergence stability?
3. Have you evaluated the experiments over multiple runs and reported the mean and standard deviation to ensure statistical robustness?
4. Could you include pseudocode or an algorithm outline (for example, in the appendix) to make the training and isotonic regression procedure clearer?

---

> ### Author Response · Authors · 2025-11-17
>
> Thank you for your review. As none of the four reviewers liked the paper, we are going to withdraw it and resubmit elsewhere. Still, here are some brief replies.
>
> Thanks for mentioning the AISTATS paper on StableMDS! We will certainly add this citation. Indeed it also suggests a stochastic solver for metric MDS; note that it does not cite _any_ of the prior _three_ papers we list in our Related Work as developing stochastic metric MDS. So this will be the fourth paper in our list. (While that paper emphasizes speed, the largest dataset it embeds has $n=4720$, which is not a lot.) We do **not** claim that our stochastic implementation of metric MDS is better than any of these four. Our focus is on non-metric MDS.
>
> Thanks for the correction about SMACOF runtime complexity. Our method is linear in the sample size, same as the previous stochastic metric MDS implementations.
>
> We will include the pseudocode and/or the full implementation into the revision.
>
> Just to clarify, the main contributions of our paper are (1) stochastic implementation of non-metric MDS; (2) simulated and real-life examples of datasets where non-metric MDS strongly outperforms metric MDS in terms of global rank-based structure. We are somewhat surprised that none of the reviewers found this aspect interesting.

---

### Note · Authors · 2025-11-25

I have read and agree with the venue's withdrawal policy on behalf of myself and my co-authors.